# Antioxidant, Cytotoxic, and Antimicrobial Activities of *Glycyrrhiza glabra* L., *Paeonia lactiflora* Pall., and *Eriobotrya japonica* (Thunb.) Lindl. Extracts

**DOI:** 10.3390/medicines6020043

**Published:** 2019-03-30

**Authors:** Jun-Xian Zhou, Markus Santhosh Braun, Pille Wetterauer, Bernhard Wetterauer, Michael Wink

**Affiliations:** Institute of Pharmacy and Molecular Biotechnology, Heidelberg University, Im Neuenheimer Feld 364, 69120 Heidelberg, Germany; junxian.zhou@stud.uni-heidelberg.de (J.-X.Z.); m.braun@uni-heidelberg.de (M.S.B.); p.wetterauer@uni-heidelberg.de (P.W.); bernhard.wetterauer@urz.uni-heidelberg.de (B.W.)

**Keywords:** TCM, phytochemistry, LC-MS/MS, antioxidant activity, ABTS, DPPH, FRAP, ascorbic acid, EGCG, total phenolics, antimicrobial activity

## Abstract

**Background:** The phytochemical composition, antioxidant, cytotoxic, and antimicrobial activities of a methanol extract from *Glycyrrhiza glabra* L. (Ge), a 50% ethanol (in water) extract from *Paeonia lactiflora* Pall. (Pe), and a 96% ethanol extract from *Eriobotrya japonica* (Thunb.) Lindl. (Ue) were investigated. **Methods:** The phytochemical profiles of the extracts were analyzed by LC-MS/MS. Antioxidant activity was evaluated by scavenging 2,2-diphenyl-1-picrylhydrazyl (DPPH) and 2,2′-azino-bis (3-ethylbenzothiazoline-6-sulphonic acid) (ABTS) radicals and reducing ferric complexes, and the total phenolic content was tested with the Folin–Ciocalteu method. Cytotoxicity was determined with a 3-(4,5-dimethylthiazol-2-yl)-2,5-diphenyltetrazolium bromide (MTT) assay in murine macrophage RAW 264.7 cells. Antimicrobial activity of the three plant extracts was investigated against six bacterial strains with the broth microdilution method. **Results:** Only Pe showed high antioxidant activities compared to the positive controls ascorbic acid and (−)-epigallocatechin gallate (EGCG) in DPPH assay; and generally the antioxidant activity order was ascorbic acid or EGCG > Pe > Ue > Ge. The three plant extracts did not show strong cytotoxicity against RAW 264.7 cells after 24 h treatment with IC_50_ values above 60.53 ± 4.03 μg/mL. Ue was not toxic against the six tested bacterial strains, with minimal inhibitory concentration (MIC) values above 5 mg/mL. Ge showed medium antibacterial activity against *Acinetobacter bohemicus*, *Kocuria kristinae*, *Micrococcus luteus*, *Staphylococcus auricularis*, and *Bacillus megaterium* with MICs between 0.31 and 1.25 mg/mL. Pe inhibited the growth of *Acinetobacter bohemicus*, *Micrococcus luteus*, and *Bacillus megaterium* at a MIC of 0.08 mg/mL. **Conclusions:** The three extracts were low-cytotoxic, but Pe exhibited effective DPPH radical scavenging ability and good antibacterial activity; Ue did not show antioxidant or antibacterial activity; Ge had no antioxidant potential, but medium antibacterial ability against five bacteria strains. Pe and Ge could be further studied for their potential to be developed as antioxidant or antibacterial candidates.

## 1. Introduction

Traditional Chinese medicine (TCM) has a history of thousands of years in China. The first professional TCM book was Shen Nong’s Chinese Materia Medica written in the Eastern Han Dynasty (AD 25–220), but before that time, people already had records of plants used as medicines. With time and the development of their practical uses, the types of traditional medicines and books about them increased gradually. TCM and the secondary metabolites of TCM plants, such as the anti-malarial drug artemisinin, have been used to treat various diseases and become more and more popular in the world, based on modern pharmacological studies [1]. In vitro antioxidant activities of extracts of TCM plants have been widely studied and the strong antioxidant activity of many TCM plants has been found to be due to high phenolic contents (flavonoids, phenolic acids, lignans, tannins, coumarins, etc.) [2,3].

Aerobic metabolism is important for most cells to produce energy. This process generates free oxygen radicals or reactive oxygen species (ROS). Excessive generation of ROS may lead to oxidative chain reactions and thus an imbalance of oxidants and antioxidants in the body, and can cause molecular damage and several health conditions [4,5,6]. For example, ROS can oxidize the purine base guanosine, leading to 8-oxoguanosine; if not repaired, this transformation can lead to mutations and proteins with impaired functions. Antioxidants (present in cells or acquired via food or medicinal plants) can delay or inhibit the oxidative reactions or scavenge initiating radicals, thus limiting the oxidative damage [7,8]. The role of antioxidants can be determined by their interaction with oxidative free radicals [9]. A diversity of antioxidants is produced in plants, and phenolics constitute a major antioxidant group in many medicinal and food plants [10,11].

*Glycyrrhiza glabra* is one of the most frequently used traditional medicine in China and Europe since long ago [12,13]; *Paeonia lactiflora* is often used together with *G. glabra* to enhance the therapeutic effect, for example, in the prescription “Shaoyaogancaotang” [14,15]; one of the main secondary metabolite in *Eriobotrya japonica*, the triterpene ursolic acid, has a similar structure as 18β glycyrrhetinic acid, which is a major secondary metabolite in *G. glabra*. So, the three plant extracts were studied in order to compare their pharmacological effects. The species have been introduced before in Reference [16]. 

A methanol extract of *Glycyrrhiza glabra* (Ge), a 50% ethanol (in water) extract of *Paeonia lactiflora* (Pe), and a 96% ethanol extract of *Eriobotrya japonica* (Ue) were studied for their antioxidant activity as well as cytotoxicity and antimicrobial activity. The solvents for the extraction were optimized according to the literature [17,18,19]. Few studies on the antioxidant activity of these species had been conducted, or only one kind of antioxidant assay was applied; and it would also be interesting to study the antibacterial capacity of the species on different bacteria to broaden their future application. Our study may help to evaluate the therapeutic potential of the species.

The phytochemical composition of the extracts was studied by LC-MS/MS and largely confirmed from other laboratories (shown in Appendix A). We employed three different assays (DPPH, ABTS, and Ferric Reduction Antioxidant Potential (FRAP) assay) to examine the potential antioxidant activity of the three plant extracts, the Folin–Ciocalteu method to determine the total phenolic contents, and a MTT assay to determine a possible cytotoxicity against murine macrophage RAW 264.7 cells. Antimicrobial activity of the three plant extracts against gram-negative (*E. coli* XL1-Blue MRF′, *Acinetobacter bohemicus*) and gram-positive (*Kocuria kristinae, Micrococcus luteus, Staphylococcus auricularis,* and *Bacillus megaterium*) bacteria was analyzed using standard broth microdilution assays.

## 2. Materials and Methods 

### 2.1. Plant Materials and Plant Extraction

The origins of the three TCM plants, and the extraction processes of Ge, Pe, and Ue have previously been described [16].

### 2.2. Reagents and Chemicals

Ascorbic acid and ferric chloride were purchased from AppliChem (Darmstadt, Germany), and gallic acid from Ferak Berlin (Berlin, Germany). Formic acid and Folin–Ciocalteu were obtained from Merck (Darmstadt, Germany), ampicillin from Panreac AppliChem (Darmstadt, Germany), and acetonitrile, 2,2-diphenyl-1-picrylhydrazyl (DPPH), 2,4,6-tris(2-pyridyl)-s-triazine (TPTZ), 2,2′-azino-bis (3-ethylbenzothiazoline-6-sulphonic acid) (ABTS), Trolox, (−)-epigallocatechin gallate (EGCG), doxorubicin, ciprofloxacin, and 3-(4,5-dimethylthiazol-2-yl)-2,5-diphenyltetrazolium bromide (MTT) from Sigma-Aldrich (Darmstadt, Germany). The reference substances paeoniflorin and ursolic acid were obtained from Baoji Herbest Bio-Tech (Baoji, China).

### 2.3. Cell Lines and Bacterial Strains

Murine macrophage cell line RAW 264.7 was a gift from PD Dr. Katharina Kubatzky (Medical Microbiology and Hygiene, Heidelberg University, Heidelberg, Germany). *Acinetobacter bohemicus* DSM 102855, *Kocuria kristinae* DSM 20032 (formerly known as *Micrococcus kristinae*), *Micrococcus luteus* DSM 20030 (synonym *Micrococcus lysodeikticus*), *Staphylococcus auricularis* DSM 20609, and *Bacillus megaterium* DSM 32 were purchased from the German Collection of Microorganisms and Cell Cultures (DSMZ, Braunschweig, Germany). *E. coli* XL1-Blue MRF′ is a cloning strain from Stratagene (Heidelberg, Germany).

### 2.4. LC-MS/MS Analysis

For Ge, the LC-MS/MS analysis was performed on a Thermo Finnigan LCQ Advantage ion trap mass spectrometer (Thermo Finnigan, San Jose, CA, USA) with an ESI source, coupled to a Thermo Scientific Accela HPLC system (MS pump plus, autosampler, and PDA detector plus) (Thermo, San Jose, CA, USA) with an EC 150/2 Nucleodur 100-3 C18ec column (Macherey-Nagel, Düren, Germany). A gradient of water and acetonitrile (ACN) with 0.1% formic acid each for ESI+ and ESI-mode was applied from 20% to 80% ACN in 20 min at 20 °C. The flow rate was 0.3 mL/min. The injection volume was about 25 µL. The MS was operated with a capillary voltage of 10 V (ESI+) or -10 V (ESI-), source temperature of 240 °C, and high purity nitrogen as a sheath and auxiliary gas at a flow rate of 70 and 10 (arbitrary units), respectively.

For Pe and Ue, the LC-MS/MS analysis was performed on a Finnigan LCQ-Duo ion trap mass spectrometer with an ESI source (ThermoQuest, San Jose, CA, USA), coupled to a Thermo Scientific Accela HPLC system (MS pump plus, autosampler, and PDA detector plus) (Thermo, San Jose, CA, USA) with an EC 150/3 Nucleodur 100-3 C18ec column (Macherey-Nagel, Düren, Germany). A gradient of water and ACN with 0.1% formic acid each was applied for Pe from 5% to 40% ACN in 100 min at 30 °C and for Ue from 5% to 80% ACN in 60 min and to 95% in another 30 min at 30 °C. The flow rate was 0.5 mL/min. The injection volume was about 20 µL. The MS was operated with a capillary voltage of 10 V (ESI+) or -10 V (ESI-), source temperature of 240 °C, and high purity nitrogen as a sheath and auxiliary gas at a flow rate of 80 and 40 (arbitrary units), respectively.

In all measurements, the ions were detected in a mass range of 50–2000 m/z. A collision energy of 35% was used in MS/MS for fragmentation. Data acquisitions and analyses were executed by XcaliburTM 2.0.7 software (Thermo Scientific, Karlsruhe, Germany). For compound determination in Ge, the positive and negative modes were used, and for Pe and Ue only the negative mode.

### 2.5. DPPH Radical Scavenging Assay

The stable free radical DPPH•, shows a deep violet color in solutions and has a strong absorption at 517 nm. When an odd electron is paired off by an antioxidant, the deep violet color disappears. The decrease in absorption is a measure for antioxidant activity [20]. The procedure was modified from Brand-Williams et al. [21]. In a 96-well plate, 100 μL of 0.2 mM DPPH• in methanol was added to 100 μL serial-diluted plant extracts and allowed to react for 30 min in darkness at ambient temperature. Ascorbic acid and EGCG were used as positive controls. The absorption was read spectrophotometrically at 517 nm with a Tecan Nano Quant infinite M200 PRO Plate Reader (Tecan, Männedorf, Switzerland). Results are expressed as EC_50_ (the concentration where 50% of the DPPH radical is inhibited). The calculation equation is: % inhibition = (AB − AE)/AB × 100
where AB and AE are the absorptions in the absence and presence, respectively, of antioxidant substances (plant extracts).

### 2.6. Assay of Trolox-Equivalent Antioxidant Capacity (TEAC)/ABTS assay 

The ABTS radical (ABTS+•) shows a blue-green color and displays absorption at 734 nm. When a pre-formed free radical ABTS+• reacts with electrons donated by an antioxidant, the color and absorption are decreased and compared with that of the standard antioxidant compound Trolox, a water-soluble vitamin E analog [22]. The procedure is according to Pietta et al. [23]. In total, 7 mM ABTS was mixed with 2.45 mM potassium persulfate in de-ionized water and the mixture was put in darkness at ambient temperature for 12–16 h to make the ABTS+• stock solution. The ABTS+• stock solution was diluted with water to obtain the working solution, which should have an absorption of 0.7 (± 0.02) at 734 nm. In 96-well plates, 250 μL of ABTS+• working solution was added to 50 μL serial-diluted plant extracts or Trolox. Trolox (0–40 μM) in 100% ethanol was used to make a standard curve. The plates were incubated at 37 °C in darkness for 6 min and the absorption was read at 734 nm with Tecan Nano Quant infinite M200 PRO Plate Reader. Ascorbic acid and EGCG were used as positive controls. Results were compared with Trolox and expressed as TEAC (Trolox equivalents in mM Trolox/mM test substance). 

### 2.7. Assay of the Ferric Reduction Antioxidant Potential (FRAP)

In the FRAP assay, the trivalent ferric ion complex (Fe^3+^ - TPTZ) is reduced by reducing agents or antioxidants under acidic conditions, to a complex of divalent ferrous ion (Fe^2+^ - TPTZ), which shows a blue color and has a peak of absorption at 593 nm [24]. The procedure was performed according to Benzie et al. [25]. Briefly, the FRAP reagent was prepared by mixing 10 mM of TPTZ in 40 mM of hydrogen chloride, 300 mM of acetate buffer (pH 3.6), and 20 mM of ferric chloride in water at a ratio of 1:10:1. In 96-well plates, 175 μL FRAP reagent solution was added to 25 μL serial-diluted substances or ferrous sulfate standards in water. The plates were incubated at 37 °C in darkness for 7 min and the absorption was measured at 593 nm with Tecan Nano Quant infinite M200 PRO Plate Reader. The results are expressed by comparison with the standard ferrous ion to obtain the ferrous equivalent, FE (mmol Fe^2+^/g test substance).

### 2.8. Total Phenolic Content Tested by the Folin–Ciocalteu Method

The colorimetric Folin–Ciocalteu method was modified from Swain et al. [26]. The final product from the reaction of the Folin–Ciocalteu method shows a blue color and can be recorded at 750 nm [27]. In total, 100 μL of Folin–Ciocalteu reagent was added to 20 μL of the plant extracts and the standard gallic acid in methanol in a 96-well plate. After 5 min, 80 μL of 7.5% sodium carbonate was added to each well. The plate was allowed to stand in darkness at ambient temperature for 2 h before the absorption was read at 750 nm with Tecan Nano Quant infinite M200 PRO Plate Reader. The standard curve was made with gallic acid (final concentration 0–40 μg/mL). The total phenolic content was compared with gallic acid to obtain the GAE (gallic acid equivalents in mg gallic acid/g test substance).

### 2.9. Cell Culture and Cytotoxicity Assay 

RAW 264.7 cells were cultured in DMEM supplemented with 10% FBS, 100 U/mL penicillin-streptomycin and 2 mM L-glutamine, and incubated at 37 °C with 5% CO_2_. The MTT assay was modified from Mosmann [28]. A density of 6 × 10^4^ RAW 264.7 cells was seeded in a 96-well plate and incubated at 37 °C for 24 h. Different concentrations of a substance dissolved in media were added to the cells for an 24 h incubation. The media were removed, and media containing 0.5% MTT were added into every well and further incubated for 2–4 h at 37 °C. Finally, after centrifuging the plate at 400 rpm for 10 min, the absorption was read at 570 nm with the Tecan Nano Quant infinite M200 PRO Plate Reader. The chemotherapeutic agent doxorubicin was used as a positive control.

### 2.10. Determination of Minimum Inhibitory Concentrations (MIC) and Minimum Bactericidal Concentrations (MBC) by Broth Microdilutions

Broth microdilution was carried out in accordance with CLSI [29]. The plant extract was dissolved in DMSO and then serial diluted with MHB from 10 mg/mL to 0.0048 mg/mL in triplicate in a 96-well plate. The final concentration of DMSO in the test did not exceed 5%. The bacterial suspensions were added to the plate to yield 5 × 10^5^ cfu/mL. The plates were incubated at 37 °C for 20 h. The lowest concentration of plant extract in the well with no visible turbidity was considered the MIC. To determine the minimum bactericidal concentration, 3 μL of suspensions from the clear wells were spread out on an LB agar plate and incubated at 37 °C until sufficient growth was obtained. The lowest concentration that reduced the number of viable cells of the initial inoculum to <0.1% was regarded as the MBC. MHB media, 5% DMSO, ampicillin, ciprofloxacin, and bacterial suspensions were used as controls, respectively.

### 2.11. Statistical Analysis

Data analysis was carried out with GraphPad Prism 6 (Graphpad Software, San Diego, CA, USA), and SigmaPlot® 11.0 (Systat Software, San Jose, CA, USA). Results were expressed as the mean ± SD. Statistical significance was evaluated using t-test and significance was set at *p* < 0.05. All experiments were performed independently at least three times.

## 3. Results

### 3.1. LC-MS/MS Analysis of Glycyrrhiza glabra Extract

As we can see from Figure 1 and Table 1, several secondary metabolites have been identified by LC-MS/MS analysis from *Glycyrrhiza glabra*, among which glycyrrhizic acid and (iso)liquiritin apioside isomers are the most abundant compounds.

### 3.2. LC-MS/MS Analysis of Paeonia lactiflora Extract

As shown in Figure 2 and Table 2, several compounds have been identified in the *Paeonia lactiflora* extract by LC-MS/MS analysis. Among them, paeoniflorin, galloylpaeoniflorin isomer, probably oxypaeoniflorin, and compounds related to pentagalloyl glucose and to benzoyloxypaeoniflorin are the abundant secondary metabolites.

### 3.3. LC-MS/MS Analysis of Eriobotrya japonica Extract

As shown in Figure 3 and Table 3, several main compounds (ursolic acid and nerolidol-trirhamnopyranosyl-glucopyranoside or loquatifolin A or 6,7-trans-nerolidol-trirhamnopyranosyl-glucopyranoside, etc.,) have been identified in the *Eriobotrya japonica* extract by LC-MS/MS analysis.

### 3.4. Antioxidant Activities and Total Phenolic Contents

The antioxidant activity of the three plant extracts was determined and compared with the known antioxidants ascorbic acid and EGCG. Results are shown in Table 4. The standard curve of Trolox in the ABTS test, ferrous sulfate in FRAP, and gallic acid equivalents in the total phenol test are provided in Appendix A. The positive control ascorbic acid showed the lowest EC_50_, i.e., the highest scavenging effect in DPPH assay. Pe had an EC_50_ value close to ascorbic acid, but slightly higher, meaning it had a slightly weaker antioxidant effect than ascorbic acid; however, this effect was not significant in ABTS and FRAP assays. Ge and Ue did not show stark antioxidant capacity in the three assays. The total phenolic contents analyzed by the Folin–Ciocalteu method are shown as GAE (the phenolic content in 1 g dried sample is equivalent to the amount of gallic acid in mg). The more phenolics in the plant extract, the stronger its antioxidant activity. Pe contained more total phenolics than the Ue and Ge.

### 3.5. Cytotoxicity

The potential cytotoxicity (IC_50_ values) of the three plant extracts in RAW 264.7 cells were assessed. The three plant extracts showed concentration-dependent inhibition of cell growth (data not shown), and they were not cytotoxic (with IC_50 values_ between 60 and 100 μg/mL) compared to the positive control doxorubicin and EGCG.

### 3.6. Antimicrobial Activity

The MIC and MBC of the three plant extracts against two gram-negative (*E. coli* XL1-Blue MRF′ and *Acinetobacter bohemicus*) and four gram-positive bacteria (*Kocuria kristinae*, *Micrococcus luteus*, *Staphylococcus auricularis,* and *Bacillus megaterium*) are presented in Table 5. At the concentrations tested, the three plant extracts varied considerably in their antimicrobial activity against the six bacterial strains. Ue restrained the growth of bacteria at or above 5 or 10 mg/mL. Ge showed intermediate antibacterial activity against all bacterial species (MIC between 0.31 mg/mL and 1.25 mg/mL), except *E. coli* XL1-Blue MRF′ (MIC > 10 mg/mL). On the other hand, Pe inhibited the growth of *Acinetobacter bohemicus*, *Micrococcus luteus,* and *Bacillus megaterium* at 0.08 mg/mL (Table 5). The control groups (5% DMSO and bacterial suspensions) showed normal bacterial growth, meaning that the solvents did not inhibit bacterial growth in any case.

## 4. Discussion

Pe has a relatively high DPPH• scavenging activity, which is comparable to that of ascorbic acid. This finding is in agreement with Lee et al. and Bae et al. [47,48]. Ge and Ue extracts were weaker antioxidants in DPPH, ABTS, and FRAP assays. The results of Ge are in agreement with literature data, in which the main plant secondary metabolites (PSMs) liquiritin, glycyrrhizin, and glycyrrhetinic acid did not scavenge the DPPH• or the effects were not strong [49,50,51]. We also tested the effect of glycyrrhizin on scavenging DPPH•, but the effect was negligible (data not shown). 

Plant polyphenols usually exhibit good antioxidant properties [11,52,53,54,55]. In detail, the specific structure of polyphenols enables them to donate hydrogen, delocalize electrons, quench singlet oxygen, and react with free radicals [56,57]. *E. japonica,* the leaves of which contain polyphenols, was found to possess a high degree of antioxidant activity and the radical scavenging activity of its seed extract increased with the polyphenol content [58,59]. The difference of the antioxidant activity of *E. japonica* between our results and the literature may be due to different solvents of the plant extracts. The three methods used employ different mechanisms, but the reducing capacity of the three plant extracts showed the same trend in the three assays (ascorbic acid or EGCG > Pe > Ue > Ge). Pe contained the most phenolics and therefore showed probably the strongest antioxidant activity.

Our previous study examined the cytotoxicity of the three plant extracts in the drug-resistant cancer cell line CEM/ADR 5000 and Caco-2 compared to the sensitive cancer cell line CCRF-CEM and HCT-116. The three plant extracts did not show strong cytotoxicity compared to the positive control doxorubicin in sensitive and resistant cell lines [16]. This time, we showed that the three plant extracts were not cytotoxic against a murine macrophage RAW 264.7 cell line, either. These results verify that the traditional usage of these plants is safe and pave the way for their future usage.

Plant extracts and essential oils have been widely studied and used as antimicrobial agents in the last decades [11,60]. The MBC of the three plant extracts is usually two to four times that of MIC, suggesting a dose-dependent effect on bacteria. The ratio of some MBC to MIC is >4, suggesting the bacteriostatic effect of the plant extracts on the bacteria. Few antimicrobial studies of the leaf extract of *E. japonica* have been conducted and it did not show toxicity against the six bacterial strains here. The *G. glabra* extract showed antimicrobial activity in some other bacterial strains [61,62,63,64,65] and medium effect against five bacterial strains in this study. The *P. lactiflora* extract was reported to exhibit antibacterial and antiviral activity [66,67]; its antibacterial effect was strong on some bacteria species here. The secondary metabolites in plants, such as saponins, phenolic compounds (e.g., flavonoids or tannins), essential oils, and monoterpenes, contribute to their antimicrobial capacity [11,68,69,70]. Wang reviewed the finding that one triterpene (18β-glycyrrhetinic acid) and four flavonoids (licochalcone A, licochalcone E, glabridin, and liquiritigenin) underlie the antimicrobial activity in *G. glabra* [71]. Low concentrations of the PSMs in *G. glabra*, such as glycyrrhizic acid, 18β-glycyrrhetinic acid, liquiritigenin, and isoliquiritigenin, were also tested in the six species, but the effect was not significant (data not shown), suggesting that these PSMs did not contribute to the medium antibacterial activity of *G. glabra*. However, polyphenols can interact with proteins in cells, because they possess several phenolic OH groups, which allow them to make hydrogen and ionic bonds with amino groups in proteins. When important bacterial proteins are affected, an antimicrobial effect can occur [72]. The strong antibacterial effect of Pe was probably due to its phenolic content. Correspondingly, the same principle might also explain the antibacterial activity of Ge and Ue. The mechanism of antimicrobial activity of *P. lactiflora* root and *E. japonica* leaves was reported to be disruption of protein and cell-wall synthesis [73]. More studies are needed to elucidate the potential of the three plant extracts as antimicrobial agents and the possible mechanisms.

## 5. Conclusions

Our results show that the three extracts of TCM plants are low-toxic, but biologically active, which would explain their wide usage in traditional medicine. Especially Pe and Ge, should be studied further for their potential to be developed as antioxidant food supplements or antibacterial drugs.

## Figures and Tables

**Figure 1 medicines-06-00043-f001:**
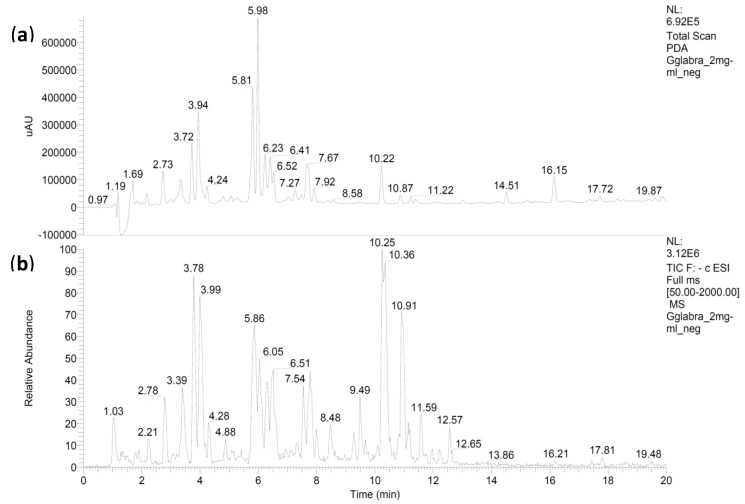
The photodiode array chromatogram (PDA) (**a**) and the total ion current (TIC) (**b**) of the *G. glabra* extract. The compounds listed in Table 1 correspond to the retention times of the TIC.

**Figure 2 medicines-06-00043-f002:**
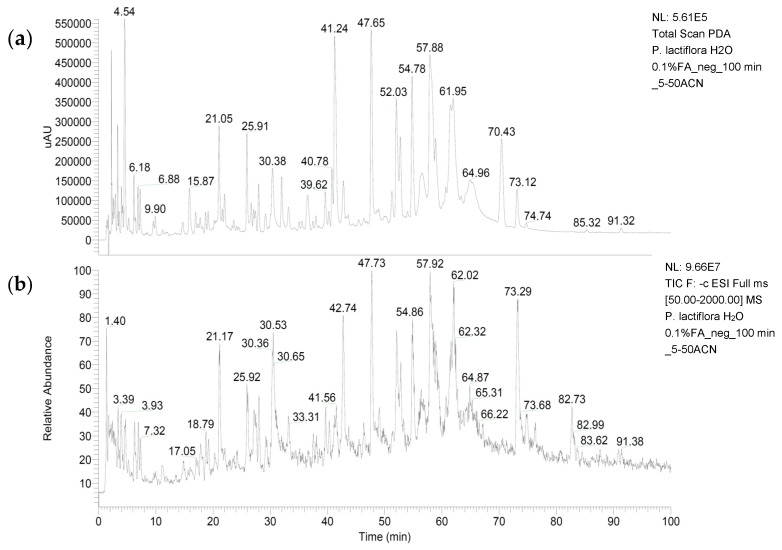
The photodiode array chromatogram (PDA) (**a**) and the total ion current (TIC) (**b**) of the *P. lactiflora* extract. The compounds listed in Table 2 correspond to the retention times of the TIC.

**Figure 3 medicines-06-00043-f003:**
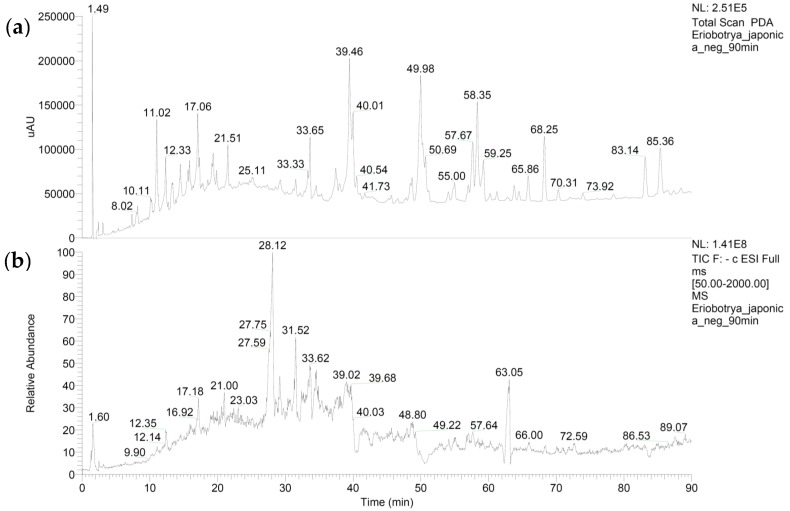
The photodiode array chromatogram (PDA) (**a**) and the total ion current (TIC) (**b**) of the *E. japonica*. The compounds listed in Table 3 correspond to the retention times of the TIC.

**Table 1 medicines-06-00043-t001:** Retention times (RT), MS data, and tentatively identified compounds in the *G. glabra* extract.

RT (min)	[M-H]^−^ (m/z)	MS/MS (m/z) from [M-H]^−^	[M-H]^+^ m/z	MS/MS (m/z) from [M+H]^+^	PDA λ_max_ [nm]	Tentative Identification	References
2.78	563.29	353.19; **443.13**; **473.12**; 503.16	565.07	409.11; **427.05**; 445.06; 457.03; 481.05; **499.03**; **511.04**; **528.96**; **546.95**	217; 274; 334	rhamnoliquiritin	[30]
3.39	577.31	353.21; 383.21; 439.18; 457.14; 473.19; 503.09; 559.26	579.18	423.07; 441.03; **525.04**; **542.95**; **560.91**	217; 272; 331	isoviolanthin	[30,31]
3.78	549.59	**255.12**; 297.14; **429.11**	551.11	388.11	217; 270; 314	liquiritin apioside isomer	[30,32]
			257.21	136.97; 146.94; 238.98		source fragment	
			419.13	**256.94**; 296.73; 364.73; 399.87		source fragment	
3.99	549.43	**255.09**; 297.17; 417.23; **429.12**	551.16	313.29; 388.11	227; 276; 311	liquiritin apioside isomer	[30,32]
			419.05	257.01		source fragment	
			257.21	136.96; 147.03; 238.97		source fragment	
4.28	417.26	n.d.	418.94	257.01	217; 276; 309	liquiritin	[30,32]
			257.17	136.96; 147.03; 238.97		source fragment	
5.86	549.43	n.d.	551.13	n.d.	219; 360	isoliquiritin apioside isomer	[30,32]
6.05	549.36	n.d.	551.14	n.d.	220; 365; 380	isoliquiritin apioside isomer	[30,32]
6.28	695.42	531.22; 549.17	431.13	269.10	218; 262; 307	licorice glycoside isomer	[30]
6.51	695.33	531.18; 549.18	697.26	668.04	220; 282; 316	licorice glycoside isomer	[30]
	725.29	255.32; 416.95; **531.21**; **549.18**	727.21	549.09; 726.16		licorice glycoside isomer	[30]
6.59	417.16	255.09	419.15	257.01	217; 297; 371	isoliquiritin	[30,32]
7.54	983.63	820.91	985.43	n.d.	216	licorice saponin A3	[30,32,33]
7.78	695.30	n.d.	697.1	516.49	218; 325; 362	licorice glycoside isomer	[30]
	725.28	n.d.	727.03	549.09; 726.16		licorice glycoside isomer	[30]
9.49	837.52	530.89; 661.45	839.24	663.02; 761.61	215	licorice saponin G2	[30,32,33]
	1675.47	837.42	-	-		dimer	
10.25	821.75	351.07; 759.49	823.31	n.d.	217; 250	glycyrrhizic acid	[30,32,33]
	1643.72	821.6	1645.69	n.d.		dimer	
10.91	821.67	n.d.	823.31	n.d.	217; 245	probably saponin	pers. com. PW
11.19	821.54	351.12	823.13	n.d.	216; 369	probably saponin	pers. com. PW
11.59	822.91	**351.06**; 646.57; 803.97	825.15	n.d.	216	licorice saponin J2	[30,32,33]
16.17	407.15	n.d.	409.07	203.03; 204.98; 247.05; 363.06; 391.00	217; 280	3-hydroxyglabrol	[30]
17.81	391.28	n.d.	393.08	204.97; 337.00	216; 282	glabrol	[30]
	782.92	n.d.				dimer	

n.d. not detectable. Fragments shown in bold are the main fragments.

**Table 2 medicines-06-00043-t002:** Retention times (RT), MS data, and tentatively identified compounds in the *P. lactiflora* extract.

RT (min)	[M-H]^−^ (m/z)	MS/MS (m/z)	PDA λmax (nm)	Tentative Identification	References
4.63	169.11	**125.22**; 126.38; 169.12	250	gallic acid	[34]
	338.62	**169.12**; 253.11; 291.91; 320.43; 339.05		gallic acid [2M-CO_2_-H]^−^	pers. com. BW
6.27	493.26*	169.18; 241.14; 283.26; 313.13; **331.04**; **403.13**	230; 273	galloylsucrose isomer	[35]
	986.96	Nl			
6.97	493.24*	211.22; **271.12**; 313.16; **331.04**; 384.22; 433.14; 475.58	230; 273	galloylsucrose isomer	[35]
	986.98	Nl			
7.32	493.19*	169.25; 271.45; **313.17**; 331.04; 389.98; 449.02	230; 274	galloylsucrose isomer	[35]
	986.88	Nl			
9.68	483.19	150.90; **169.14**; 193.34; 223.18; 271.10; 295.22; **313.12**; **331.11**; 426.15	230; 273	digalloyl glucose	[36]
21.17	495.22	177.12; 299.13; **333.19**; 387.11; 447;06; **465.17**; 477.11	227; 253	probably oxypaeoniflorin	[37]
27.50	525.03	196.34; 213.42; 283.35; 317.24; 357.38; 391.56; 435.70; 475.77; **479.07**; 524.58	221; 237; 273	albiflorin	[38]
29.28	197.17	124.32; 141.56; 153.01; **169.18**; **197.16**	231; 272	probably ethyl gallate	[36]
	394.72	-			
30.14	635.1	207.31; 234.79; 313.09; 358.75; **465.13**; 483.14; 524.23; 566.84; 589.17	233; 276	trigalloyl glucose	[36]
30.53	449.04	**165.01**; 179.34; 205.10; 261.31; 282.87; 309.03; **326.95**; 398.60; 431.13	243; 274	peaoniflorin [M-CH_2_O-H]^−^	Standard
	479.04	149.09; 177,10; 248.83; 267.08; 309.08; **326.98**; 355.61; **356.96**; 432.93; **449.16**; 460.71; 477.94		paeoniflorin [M-H]^−^	[36,39]
	525.01	176.88; 282.89; 327.09; 356.85; **449.01**; 476.31; **478.83**; 494.01; 506.96		paeoniflorin [M+HCOOH-H]^−^	[37,38]
32.08	463.24	**301.30**; 343.04; 394.94; 445.33; 463.25	253; 361; 280	visculdulin I 2´-glycoside	[38]
39.69	787.17**	295.17; 447.22; 465.33; 483.29; **617.32**; **635.17**	232; 277	probably tetragalloyl glucose isomer	[36]
40.29	611.22	301.30; **343.35**; 385.33; 427.35; **445.21**	232; 272		
40.93	477.22	160.70; 300.45; 315.12; 357.02; 408.88	227; 253; 360	probably related to isorhamnetin 7-*O*-glucoside	[36]
41.36	301.31	145.14; 185.44; 229.47; 257.47; **301.33**	249; 367		
	509.10	202.99; 254.25; 314.06; 372.80; 440.82; **463.22**; 480.12			
	787.13**	295.23; 403.40; 465.43; 530.46; 573.46; **617.18**; **635.14**; 679.31; 719.88		probably tetragalloyl glucose isomer	[36]
42.74	631.25	271.16; 313.23; 399.30; 465.30; 479.28; 491.23; 509.22; 585.17; **613.18**	234; 274	gallylpaeoniflorin isomer	[35,36,37,39,40]
47.77	939.11	277.04; 341.21; 385.21; 447.13; 511.35; 573.25; 599.15; **617.19**; 725.13; **769.12**; **787.03**	234; 269	probably related to pentagalloyl glucose	[36]
48.49	615.18	239.29; 263.04; **281.22**; 401.27; **431.23**; **447.22**; 459.26; **477.22**; 495.21; 567.13; **585.16**; **597.17**	232; 275	mudanpioside H	[36]
61.98	599.26	241.29; 281.46; 333.31; 385.39; 403.06; 429.22; 447.51; 459.42; **477.31**; **569.17**; 581,12	233; 274	probably related to benzoyloxypaeoniflorin	[39]
	1394.92	599.23; 937.97; 970.98; **1090.82**; 1126.36; **1165.39**; **1241.78**; 1257.75; 1309.67; **1318.90**; 1336.64			
73.24	628.99	552.66; 582.88	239; 274	probably related to benzoylpaeoniflorin	[37,39]
	1212.42	876.29; 1067.88			

* isomers; ** related; nl: neutral loss. Fragments shown in bold are the main fragments.

**Table 3 medicines-06-00043-t003:** Retention times (RT), MS data, and tentatively identified compounds in the *E. japonica* extract.

RT (min)	[M-H]^−^ (m/z)	MS/MS (m/z)	PDA λmax (nm)	Tentative Identification	References
11.05	352.96	110.40; 143.67; **179.20**; **191.20**; 284.50; 312.26	234; 295; 325	probably chlorogenic acid	[41]
	420.95	259.61; **301.23**; **331.19**; 343.20; 352.64; 360.36; **375.14**; 385.20; 392.55; 403.15		n.c.	
17.18 (16.48–17.26)	463.17	151.00; 179.09; 190.34; 221.17; 255.30; 271.50; **300.25**; **301.16**; 325.03; 343.10; 373.35; 400.93; 418.54; 445.14	234; 347	hyperoside or isoquercetin isomers	[42,43]
	547.19	220.44; 292.64; **310.81**; 384.91; 437.90; 478.82; **500.58**; 515.88		n.c.	
	593.04	255.34; **284.19**; 327.19; 411.21; **429.21**; 447.18; 473.09; 565.32		n.c.	
	855.43	417.31; 545.16; **563.31**; 691.43; **709.34**; 735.38; 864.27		n.c.	
	901.14	299.82; 439.19; 610.71; 721.14; 763.80; 854.33; 914.59		n.c.	
28.12	821.37	511.74; **529.60**; 657.60; **675.39**; 721.14; 766.16	221; 234; 280	(trans)nerolidol-trirhamnopyranosyl-glucopyranoside or loquatifolin A	[44] (compound 1 or 4)
	867.12	596.27; 675.92; 690.31; 721.42; 740.66; 786.99; 815.64; 820.05; 833.88		n.c.	
	1688.98	551.69; 696.87; **719.35**; 821.28; **865.28**; 881.05; **1275.89**; 1396.00; 1541.66		n.c.	
29.11	807.37	529.33; 661.43; **675.31**	217; 234; 280	unknown new compound	[44](compound 2)
	853.02	350.34; 454.48; 503.33; 649.33; 731.12; 784.53; 809.49; 839.42; 853.53		n.c.	
31.52	675.31	204.97; 307.09; **383.17**; 467.33; **529.20**; 574.81	221; 234; 283; 312	nerolidol-dirhamnopyranosyl-glucopyranoside	[44] (compound 3 or 5)
	721.21	490.37; 597.16; **675.29**		[M+HCOO-H]^−^ of 675.31 [M-H]^−^	[44]
33.12-33.67	967.47	309.10; 351.25; 395.15;437.27; 511.25; **529.48**; 579.22; 639.41; **657.34**; **675.47**; 717.98; **743.29**; 761.35; **803.44**; **821.38**; 848.32; 865.41; 922.75; 945.31	334; 289; 323	n.c.	
	997.51	381.23; 467.03; 511.14; **529.05**; 543.22; 567.25; 603.04; **657.15**; **675.18**; 697.48; 721.08; 773.26; **803.34**; **821.36**; 833.47; 915.25; 938.24		probably nerolidol--rhamnopyranosyl -rhamnopyranosyl--(4-trans-feruloyl)-rhamnopyranosyl--glucopyranoside	[45]
	1065.21	405.44; 513.91; 579.73; 675.96; 1020.72; 1033.32; 1041.27; 1057.41; 1067.28		n.c.	
50.02	633.52	339.61; 469.47; 487.39; **513.48**; 571.44; **589.50**; 615.47; 633.50	219; 235; **310**	probably 3-*O*-*p*-coumaroyltormetic acid	[46]
	1267.27	1102.55		n.c.	
62.90	523.39	-	219; 235; 281	Usolic acid (monomer adduct)*	Standard
	933.70	408.37; **455.50**; 500.98; 584.57; 745.53; 870.99; 933.70		Usolic acid [2M + Na^+^ -2H^+^]^−^	Standard
	1411.83	455.52; 501.44; **933.71**; 1302.93; 1377.25; 1410.15		Usolic acid [3M + 2Na^+^ -3H^+^]^−^	Standard
	1885.16	934.78; 1391.11; **1406.28**; 1447.24; 1608.90; 1743.24; 1855.24		Usolic acid [4M + 3Na^+^ -4H^+^]^−^	Standard

All further mass peaks were assumed to be chlorophyll related, because of absorption maxima of 408 nm and higher. n.c.: not classifiable. Fragments shown in bold are the main fragments.* Ursolic acid shows, instead of its monomer ion (m/z 455.50 [M-H^+^]^−^), an unknown adduct combination (X) with m/z 523.39 [M+X-H^+^]^−^ and forms additional dimer, trimer, and tetramer adducts with Na^+^ ions.

**Table 4 medicines-06-00043-t004:** The in vitro antioxidant capacity and total phenolic content of the plant extracts.

Plant Extracts	DPPH EC_50_ (μg/mL)	TEAC (mM Trolox/mM)	FE (mmol Fe^2+^/g)	GAE (mg gallic acid/g)
Ascorbic acid	2.31 ± 0.01	6363.67 ± 32.37	14,268.44 ± 66.18	-
EGCG	9.20 ± 1.18	15,708.35 ± 54.72	25,318.57 ± 114.83	-
*Glycyrrhiza glabra* extract	116.17 ± 0.55	672.19 ± 5.06	477.42 ± 13.00	34.19 ± 2.07
*Paeonia lactiflora* extract	5.15 ± 0.05	2567.26 ± 32.83	3504.07 ± 51.07	323.19 ± 10.19
*Eriobotrya japonica* extract	35.50 ± 1.99	758.63 ± 5.23	1464.28 ± 8.32	131.32 ± 12.33

-: not tested; TEAC: Trolox equivalents in mM Trolox/mM test substance; FE: ferrous equivalents in mmol Fe^2+^/g test substance; GAE: gallic acid equivalents in mg gallic acid/g test substance.

**Table 5 medicines-06-00043-t005:** Minimum inhibitory concentration (MIC) and minimum bactericidal concentration (MBC) of the three plant extracts.

Bacteria	MIC MBC	Ampicillin (μg/mL)	Ciprofloxacin (μg/mL)	*Glycyrrhiza glabra* Extract (mg/mL)	*Paeonia lactiflora* Extract (mg/mL)	*Eriobotrya japonica* Extract (mg/mL)
*E. coli* XL1-Blue MRF′	MIC	8	0.03	>10	2.5	10
	MBC	16	0.06	-	5	-
*Acinetobacter bohemicus*	MIC	2	0.03	1.25	0.08	5
	MBC	8	0.05	2.5	1.25	10
*Kocuria kristinae*	MIC	0.13	0.13	0.63	1.25	>10
	MBC	0.25	0.5	1.25	2.5	-
*Micrococcus luteus*	MIC	0.25	0.5	0.31	0.08	10
	MBC	2	2	1.25	0.63	-
*Staphylococcus auricularis*	MIC	0.5	0.06	0.63	1.25	5
	MBC	4	0.13	1.25	>10	10
*Bacillus megaterium*	MIC	0.25	0.06	0.31	0.08	10
	MBC	1	0.13	0.63	0.31	-

-: not detectable.

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
