# Peer review of "Antioxidant, Cytotoxic, and Antimicrobial Activities of *Glycyrrhiza glabra* L., *Paeonia lactiflora* Pall., and *Eriobotrya japonica* (Thunb.) Lindl. Extracts"

_medicines, 2019, doi:10.3390/medicines6020043_

Round 1
Reviewer 1 Report
Review of manuscript medicines-464031.
The authors investigated antioxidant, antimicrobial and cytotoxic effects of extracts of three important medicinal plants - Glycyrrhiza glabra, Paeonia lactiflora, and Eriobotrya japonica. These plants are very popular in traditional medicine for a long time. However, the knowledge of the constituents and activities are not complete. In this context, the topic of the manuscript is timely.
The authors explored not only the bioeffects but also constituents of the plant species. The employed methods were adequate. The bioactivities were studied by well established and standard methods. This approach was important and useful because the results can be compared with the results of other authors.
The results are supported by the obtained data and they are properly discussed. Furthermore, the authors cited relevant refences. Therefore, I recommend the manuscript for publication.
Author Response
Thank you for your comment.
Reviewer 2 Report
The paper of Zhou et al. showed biological properties of three medicinal plant extracts used in Traditional Chinese Medicine performing a varied scientific protocol.
Phytochemical analyses represent the best section of the article.
Biological tests, however, were adequately performed and results well described.
Nevertheless, some major and minor pitfalls arose.
Introduction:
Please write botanical names in correct form: Paeonia lactiflora Pall., Glycyrrhiza glabra L., Eriobotrya japonica (Thunb.) Lindl.
Studied samples need to be described: in introduction authors should better describe pharmacological profile of studied species, both in TCM and in Western medicine; moreover, authors should address their aim of research with regards to known or novel biological properties of studied species and clearly stated why they chose methanolic or hydroethanolic extracts.
Microbiological tests were conducted considering few "strange" strain and it seems a test not conceived from the start, but an experimental add. I ask if other tests against more common and interesting pathogenic microbial strains such as S. aureus, S. epidermidis, S. pyogenes, S. salivarius, P. aeuginosa, P. mirabilis, A. fumigatus, K. pneumoniae, H. influenzae, H. pylori, C. albicans (just to cite the most interesting ones) were thought and done.
MIC and MBC should be described according to EUCAST guidelines and DMSO effect clearly stated.
Discussion is very poor, being a mere summary of results.
Discussion and conclusion should draw a story around the paper and give meaning to the research in order to consider it as a step forward in medicine.
I strongly suggest to reconstruct the whole story of the manuscript improving point by point the discussion of each test and results.
Author Response
Please write botanical names in correct form: Paeonia lactiflora Pall., Glycyrrhiza glabra L., Eriobotrya japonica (Thunb.) Lindl.
Done.
Studied samples need to be described: in introduction authors should better describe pharmacological profile of studied species, both in TCM and in Western medicine; moreover, authors should address their aim of research with regards to known or novel biological properties of studied species and clearly stated why they chose methanolic or hydroethanolic extracts.
The pharmacological profile of studied species was introduced in the Ref. 12.
It has been revised.
Microbiological tests were conducted considering few "strange" strain and it seems a test not conceived from the start, but an experimental add. I ask if other tests against more common and interesting pathogenic microbial strains such as S. aureus, S. epidermidis, S. pyogenes, S. salivarius, P. aeuginosa, P. mirabilis, A. fumigatus, K. pneumoniae, H. influenzae, H. pylori, C. albicans (just to cite the most interesting ones) were thought and done.
The bacteria used for our tests were selected to include the different types of cell walls (Gram-positive and Gram-negative bacteria were used) as well as various genera (six different genera were used).
Although we find merit in your suggestion to use pathogens, this is out of the scope of our present manuscript. This is the first study estimating the antimicrobial activities of Eriobotrya japonica, Glycyrrhiza glabra and Paeonia lactiflora leaf extracts using broth microdilution assays. As such, it was designed to identify possible antimicrobial effects and simultaneously pave the way for future in-depth susceptibility testing, including tests with clinically relevant strains. This was pointed out stating that more studies are needed to evaluate the antimicrobial activity of the three plant extracts on different bacterial strains and unravel their mechanisms of action.
MIC and MBC should be described according to EUCAST guidelines and DMSO effect clearly stated.
Thank you for pointing out that the effects of DMSO on the strains under study went unnoticed. It is important mentioning that DMSO did not inhibit bacterial growth. We added this information to Table 6.
We followed the method of CLSI instead of EUCAST (see method 2.10).
Discussion is very poor, being a mere summary of results.
Discussion and conclusion should draw a story around the paper and give meaning to the research in order to consider it as a step forward in medicine.
I strongly suggest to reconstruct the whole story of the manuscript improving point by point the discussion of each test and results.
Good idea. The ms has been revised.
Reviewer 3 Report
the comments were added directly to the pdf file of the manuscript
Author Response
Result 3.3
As shown in Figure 3 and Table 3, several main compounds (ursolic acid and nerolidol-trirhamnopyranosyl-glucopyranoside or loquatifolin A or 6,7-trans-nerolidol-trirhamnopyranosyl-glucopyranoside, etc.) have been identified in the Eriobotrya japonica extract by LC-MS/MS Analysis.
Table 3 see the revised ms in word
Table 5 has been removed
Table 6 now 5 The MIC and MBC values can be seen directly from the table, which is easy and even people don't need to read the text. So this table is good.
Round 2
Reviewer 2 Report
Authors actually improved the MS and I thank them for having addressed all the revisions suggested.